# Biomarkers in Fabry Disease. Implications for Clinical Diagnosis and Follow-up

**DOI:** 10.3390/jcm10081664

**Published:** 2021-04-13

**Authors:** Clara Carnicer-Cáceres, Jose Antonio Arranz-Amo, Cristina Cea-Arestin, Maria Camprodon-Gomez, David Moreno-Martinez, Sara Lucas-Del-Pozo, Marc Moltó-Abad, Ariadna Tigri-Santiña, Irene Agraz-Pamplona, Jose F Rodriguez-Palomares, Jorge Hernández-Vara, Mar Armengol-Bellapart, Mireia del-Toro-Riera, Guillem Pintos-Morell

**Affiliations:** 1Laboratory of Inborn Errors of Metabolism, Laboratoris Clínics, Vall d’Hebron Barcelona Hospital Campus, Vall d’Hebron Hospital Universitari, Passeig Vall d’Hebron 119-129, 08035 Barcelona, Spain; jaarranz@vhebron.net (J.A.A.-A.); ccea@vhebron.net (C.C.-A.); 2Department of Internal Medicine, Vall d’Hebron Barcelona Hospital Campus, Vall d’Hebron Hospital Universitari, Passeig Vall d’Hebron 119-129, 08035 Barcelona, Spain; mcamprodon@vhebron.net (M.C.-G.); david.moreno.m@icloud.com (D.M.-M.); 3Unit of Hereditary Metabolic Disorders, Vall d’Hebron Barcelona Hospital Campus, Vall d’Hebron Hospital Universitari, Passeig Vall d’Hebron 119-129, 08035 Barcelona, Spain; atigri@vhebron.net (A.T.-S.); mdeltoro@vhebron.net (M.d.-T.-R.); guillem.pintos@vhir.org (G.P.-M.); 4Lysosomal Storage Disorders Unit, Royal Free Hospital NHS Foundation Trust and University College London, London WC1E 6BT, UK; 5Neurodegenerative Diseases Laboratory, Vall d’Hebron Institut de Recerca (VHIR), Vall d’Hebron Barcelona Hospital Campus, Vall d’Hebron Hospital Universitari, Passeig Vall d’Hebron 119-129, 08035 Barcelona, Spain; sara.lucas@vhir.org (S.L.-D.-P.); jorhernandez@vhebron.net (J.H.-V.); mar.armengol@vhebron.net (M.A.-B.); 6Department of Neurology, Vall d’Hebron Barcelona Hospital Campus, Vall d’Hebron Hospital Universitari, Passeig Vall d’Hebron 119-129, 08035 Barcelona, Spain; 7Department of Clinical and Movement Neurosciences, UCL Queen Square Institute of Neurology, London WC1N 3BG, UK; 8Functional Validation & Preclinical Research, Drug Delivery & Targeting Group, CIBIM-Nanomedicine, Vall d’Hebron Institut de Recerca (VHIR), Universitat Autònoma de Barcelona (UAB), 08035 Barcelona, Spain; marc.molto@vhir.org; 9Networking Research Center on Bioengineering, Biomaterials and Nanomedicine (CIBER-BBN), 08035 Barcelona, Spain; 10Department of Nephrology, Vall d’Hebron Barcelona Hospital Campus, Vall d’Hebron Hospital Universitari, Passeig Vall d’Hebron 119-129, 08035 Barcelona, Spain; iagraz@vhebron.net; 11Department of Cardiology, Vall d’Hebron Barcelona Hospital Campus, Vall d’Hebron Hospital Universitari, Passeig Vall d’Hebron 119-129, 08035 Barcelona, Spain; jfrodriguez@vhebron.net; 12Department of Pediatric Neurology, Unit of Hereditary Metabolic Disorders, Vall d’Hebron Barcelona Hospital Campus, Vall d’Hebron Hospital Universitari, 08035 Barcelona, Spain

**Keywords:** fabry disease, classic phenotype, late-onset phenotype, biomarkers, cardiomyopathy, chronic kidney disease, vasculopathy, lyso-Gb3, Gb3, inflammatory response

## Abstract

Fabry disease (FD) is a lysosomal storage disorder caused by deficient alpha-galactosidase A activity in the lysosome due to mutations in the GLA gene, resulting in gradual accumulation of globotriaosylceramide and other derivatives in different tissues. Substrate accumulation promotes different pathogenic mechanisms in which several mediators could be implicated, inducing multiorgan lesions, mainly in the kidney, heart and nervous system, resulting in clinical manifestations of the disease. Enzyme replacement therapy was shown to delay disease progression, mainly if initiated early. However, a diagnosis in the early stages represents a clinical challenge, especially in patients with a non-classic phenotype, which prompts the search for biomarkers that help detect and predict the evolution of the disease. We have reviewed the mediators involved in different pathogenic mechanisms that were studied as potential biomarkers and can be easily incorporated into clinical practice. Some accumulation biomarkers seem to be useful to detect non-classic forms of the disease and could even improve diagnosis of female patients. The combination of such biomarkers with some response biomarkers, may be useful for early detection of organ injury. The incorporation of some biomarkers into clinical practice may increase the capacity of detection compared to that currently obtained with the established diagnostic markers and provide more information on the progression and prognosis of the disease.

## 1. Introduction

Fabry disease (OMIM-301500, FD) is a lysosomal storage disease with X-linked inheritance [1] secondary to mutations in the GLA gene, which cause absence or decreased activity of the lysosomal hydrolase a-galactosidase A (a-gal A). The consequent accumulation of its primary substrate globotriaosylceramide (Gb3) and derivatives, results in the injury of multiple organs and system dysfunction [2,3].

FD has an estimated overall incidence of between 1/40,000 and 1/117,000 live births [4,5]. However it could be underestimated, since there are no recent studies on this. Clinically, it has a wide phenotypic variability ranging from a multiorgan involvement, called classic phenotype, to pauci-symptomatic or atypical forms, known both as late-onset or non-classic phenotype. The classic form of the disease usually starts in childhood or adolescence and is characterized by neuropathic pain, gastrointestinal manifestations, angiokeratomas, hypohydrosis, exercise intolerance and cornea verticillata, progressing later in adulthood to chronic renal disease, cardiomyopathy, and cerebrovascular disease. However, late-onset forms can initiate in adulthood and commonly have exclusive renal or cardiac involvement [6,7,8,9]. Diagnosis is confirmed by measuring a-gal A activity in leukocytes and by genetic analysis. Due to X chromosome inactivation, female patients may present normal enzyme activity, thus they require confirmation by genetic analysis [10]. The diagnostic process can also be initiated from newborn screening (NBS) programs resulting in asymptomatic individuals or by screening programs in selected high-risk populations. Since a-gal A activity is normal in up to 30–50% of patients with a definitive diagnosis of Fabry disease, lyso-Gb3 could be useful to increase the detection capacity in such cases [11,12,13].

Available treatments for FD currently include intravenous enzyme replacement therapy (ERT), and oral chaperone treatment. ERT was the first available treatment and is able to reduce Gb3 accumulation in some cell types [14,15,16,17,18]. There is growing evidence that ERT treatment is most effective when initiated early in the course of the disease [19,20]. For this reason, it is important to search for early biomarkers that contribute to disease detection before irreversible changes develop. Our work aims to review the mediators involved in different pathogenic mechanisms of FD that have been studied as potential biomarkers, and that are good candidates to be incorporated into clinical laboratory practice.

The PubMed database was searched, combining appropriate terminology related to this topic by Boolean operators. Any type of study, both in humans of any age and in animal and cellular models was included. From the results found, only those studies that evaluated biomarkers that could be easily incorporated into clinical practice were included, mainly taking into account criteria of invasiveness of the sample and methodological feasibility of the tests. Finally, the biomarkers found in the filtered results have been classified into two groups, i.e., biomarkers related to the accumulation of substrate and those related to the response to the initial injury.

## 2. Pathogenesis

The pathogenic pathways that associate the accumulation of Gb3 and its deacylated derivative, lyso-Gb3, with the dysfunction of the various affected organs are not yet precisely known.

Endothelial vascular involvement is the main pathological alteration of the disease. Regarding the pathogenic mechanisms described in its development, several studies in vitro and in animal models suggest that accumulation of, or chronic exposure to, lyso-Gb3 promotes the proliferation of smooth muscle cells and contributes to the thickening of the arterial intima-media layer in patients with FD [21,22,23]. In response to this stimulatory effect, an influx of inflammatory cells to the arterial media layer occurs, as well as activation of the renin-angiotensin system, secretion of adhesion molecules and cytokines, and the increase in extracellular matrix (ECM), determining a pro-inflammatory effect on leukocytes and endothelial cells. Increased reactive oxygen species (ROS) and decreased nitric oxide (NO) also contribute to endothelial damage [24,25,26]. A different hypothesis about the development of vasculopathy, proposes that the accumulation of endothelial Gb3 causes a dysregulation of the enzyme nitric oxide endothelial synthase (eNOS) with the consequent formation of oxidant species derived from NO, which could be direct markers of vasculopathy in FD [27,28,29,30,31,32,33,34]. In addition, Gb3 could also contribute to endothelial damage from other different pathogenic mechanisms related to KCa3.1 channel dysfunction [35,36]. Accumulation of intracellular Gb3 and/or exposure of Gb3 or lyso-Gb3 to blood cell types could also activate other pathogenic pathways such as innate immunity [37,38], alteration of autophagy, or mitochondrial function [39]. Over time, this chronic inflammatory state will cause the development of fibrosis in different tissues, mainly in the kidney and heart with the consequent associated clinical manifestations.

Moreover, pathogenic mechanisms may differ depending on the cell type or exposed tissue. In renal involvement, podocyte injury caused by Gb3 or lyso-Gb3 accumulation exposure appears to be the initial step that induces different pathogenic mechanisms that may be associated with the progression of nephropathy [3,38,40,41,42,43,44]. Nevertheless, the substrate can also accumulate in other renal cell types [45,46] and activate parallel signaling cascades that contribute to the development of both glomerular sclerosis and tubulo-interstitial fibrosis. As a consequence of ischemia, these are irreversible situations that will lead to end-stage kidney disease [47,48,49,50,51,52,53]. In this regard, different studies have shown the participation of some mediators involved in such mechanisms in FD [42,43,46,54,55,56,57]. It is also important to consider especially proteinuria, since it is considered an important variable for the diagnosis and monitoring of patients with Fabry disease. Proteinuria is secondary to the loss of integrity of the glomerular basement membrane and also plays an important role in the progression of renal injury towards tubulointerstitial fibrosis [58].

With regard to cardiac involvement, accumulation of glycosphingolipids has been shown in different cell types including cardiomyocytes. The accumulation of Gb3 and derivatives in cardiomyocytes could induce an inflammatory response, increased oxidative stress and apoptosis [59,60]. Several mediators involved in the various pathogenic mechanisms such as immune response activation, proliferation and hypertrophy of cardiomyocytes, oxidative stress, and fibrosis development, have been associated with Fabry cardiomyopathy [60,61,62,63,64,65]. However, it should be noticed that systemic vasculopathy, and its derived inflammatory cascade might participate in the progression of organ injury related to cardiomyopathy and nephropathy [23,24,25,62,66].

Figure 1 and Figure 2 outline the main pathogenic pathways of the disease and several mediators involved in them.

## 3. Markers of Fabry Disease

### 3.1. Clinical Markers

Several clinical markers are available for the evaluation, follow-up, and monitoring of treatment in FD [67]. The main markers of renal function assessment are proteinuria and/or albuminuria and glomerular filtration rate. Evidence of Gb3 accumulation by renal biopsy is also considered in cases where diagnosis can be challenging and/or if there is uncertainty about whether to start treatment. The most established markers in the evaluation of cardiac function are blood pressure measurement, cardiac rhythm abnormalities, evidence of cardiac hypertrophy in echocardiography, evidence of late gadolinium enhancement (LGE), low T1 mapping and cardiac hypertrophy as assessed by cardiac magnetic resonance imaging (cMRI) [68], as well as circulating levels of troponins (Tn) and natriuretic peptides. With respect to cerebrovascular events, a cerebral MRI is the most established procedure for the diagnosis of a stroke, and may also show a pattern of white matter lesions due to small vessel alteration. Basilar artery dolichoectasia is also characteristic.

Measurement of plasma and/or urinary Gb3 levels have been used to assess the burden or global activity of the disease [67,69,70]. However, its sensitivity is limited in late-onset forms and/or in female patients, whose levels in some cases overlap with those of the healthy population [71,72], thus making it necessary to search for new biomarkers with more diagnostic sensitivity.

### 3.2. New Markers. Candidate Severity and Predictive Biomarkers in Fabry Disease

Different independent studies, in animal models, in vitro or in patients, have focused on the assessment of some mediators involved in the generation and progression of injury in the different organs. This section describes those that we consider to be of remarkable interest, as they could be potential non-invasive early markers, be easily incorporated into routine clinical practice, and help decision making regarding diagnosis and therapeutic management.

#### 3.2.1. Accumulation Biomarkers

Increased levels of lyso-Gb3 have been shown in plasma, urine or dried blood spot (DBS) in FD patients, and have less overlap between patient and control populations compared to those of the acylated analog Gb3 [71,73,74,75,76,77]. Thus, lyso-Gb3 has been shown to be a good diagnostic marker of the disease either for males [12], or for the classic phenotype [78], with a diagnostic sensitivity and specificity close to 100%. Some studies have proposed cut-off points to try to discriminate patients with atypical variants from controls [79]. However, in these variants, concentrations of plasma lyso-Gb3 are usually only slightly increased, or even normal in some females [11,12,74], making it difficult to establish a cut-off point. Nevertheless, it has recently been proposed that the a-gal A activity/lyso-Gb3 ratio in DBS could be a good diagnostic marker in females, as it has a higher detection sensitivity compared to that of the isolated measurement of lyso-Gb3 [80].

The association of plasma lyso-Gb3 levels with disease severity or with some cardiac and neurological manifestations in FD patients has been reported in some cases. For example, in one study of patients with the classic phenotype, concentration of plasma lyso-Gb3 was identified as an independent risk factor for the development of white matter lesions in males and left ventricular mass in females [71]. Other studies directed to patients with classic and atypical phenotype found associations between the concentration of plasma lyso-Gb3 and both left-ventricular mass in females [23] and ventricular hypertrophy in males and females [78]. Finally, association has also been demonstrated between the concentration of plasma lyso-Gb3 and both disease-associated cardiomyopathy [66,81], and LGE on cardiac MRIs [63]. However, other studies have not found such associations [23,82]. With regard to urinary lyso-Gb3, its levels correlate with proteinuria and albuminuria, although not with glomerular filtration rate, so this marker does not seem a good indicator of renal function [76].

Regarding the response to treatment, the concentration of plasma lyso-Gb3 tends to decrease significantly during treatment, especially in hemizygous patients but there is controversy about the possible association between changes in levels of plasma lyso-Gb3 with clinical response. While some studies have found no associations [82] others found that changes in the levels of plasma lyso-Gb3 were associated with changes in left ventricular mass, interventricular septal thickness, or with the development of white matter lesions [66,83]. Therefore, from the exposed findings, it could be deduced that the association of lyso-Gb3 levels with clinical manifestations and with the response to treatment seem limited. However, the disparity of results in such studies may be influenced by the different types of study designs, such as the composition of the cohorts or the inclusion criteria of the participants.

Several metabolomic studies show that there are different structural variants of Gb3, lyso-Gb3, and galabiosylceramide (Gb2) molecules that differ in the structure of the sphingosine group, called analogues [84,85,86,87]. The development of UHPLC-MS/MS methods has made it possible to quantify some of them in plasma and urine. Such methods have shown that levels of plasma and urinary lyso-Gb3 and some of its analogues are higher in patients compared to controls, being specifically higher in males, and these levels decrease significantly with treatment. In addition, the relative proportion of urinary excretion of analogues to that of lyso-Gb3 is higher than that observed in plasma [88,89,90,91], which makes lyso-Gb3 analogues in urine a good candidate as a disease marker, especially in children, who excrete a lower percentage of lyso-Gb3 than adults [91]. It should be noted that the excretion profile of lyso-Gb3, Gb3 and its analogues may vary depending on the causal genetic alteration [91], and it may be necessary to incorporate the full profile of lyso-Gb3 analogues in urine to expand the diagnostic information of patients.

Recently, it was proposed that the sum of urinary lyso-Gb3 levels plus that of its analogues may have diagnostic utility, as it has a sensitivity and specificity of 100% for the diagnosis of both classic and non-classic forms. It can be a good diagnostic marker, complementary to the measurement of a-gal A activity and the concentration of plasma lyso-Gb3 [92]. However, it should be noted that lyso-Gb3 and analogues profiled in urine have a high interindividual variability [89], so more studies with a larger number of patients are required to validate its clinical utility.

Levels of lyso-Gb3 analogues have also been associated with some clinical manifestations. Ferreira et al. [90] found that levels of plasma and urinary lyso-Gb3 analog (+50) were associated with a cardiac phenotype of FD. Another study noted that levels of plasma lyso-Gb3, and levels of urinary lyso-Gb3 analogues (+16, +34, +50) were associated with left ventricular mass index (LVMI) and the MSSI severity score index [93]. These results suggest that lyso-Gb3 analogues could also be potential indicators of severity and progression of cardiac injury.

Other glycosphingolipid studies have proposed the plasma and urinary levels of different Gb3 and Gb2 isoforms, as potential specific biomarkers of the disease. These isoforms result from the different lengths, double bonds, or other modifications of the fatty acid chains that compose glycosphingolipids. Heywood et al. analyzed plasma and urinary levels of different substrates of the glycosphingolipid degradation pathway in patients, and found that the urinary levels of the long-chain ceramide dihexoside (CDH) isoforms (which includes the sum of the structural isomers galabiosylceramide (Gb2) and lactosylceramide (LacCer)) were increased in patients. They appeared to have greater detection capacity in asymptomatic females compared to plasma lyso-Gb3, and were similar to that obtained with the sum of urinary levels of lyso-Gb3 plus its analogues [94]. With the chromatographic method developed by Boutin et al., levels of Gb2 were quantified in urine separately from those of its isomer LacCer, verifying that the isolated Gb2 measure increased diagnostic sensitivity in females compared to that obtained with the sum of the two isomers [95].

However, the methodology used to separate such isomers is more complex and less adaptable to the routine practice of a clinical laboratory. High concentrations of methylated Gb3 isoforms were also found in urine of patients compared to controls, including patients with atypical variants [96].

Finally, Gb3 expression levels were studied in different cell membranes (CD77). In human tubular cells, the expression of CD77 increases when the expression of a-Gal A is inhibited in vitro and it is significantly reduced when treated with agalsidase-α [97]. In peripheral blood mononuclear cells (PBMC) of FD patients, Gb3 levels were also significantly increased compared to control subjects [97,98] and they decreased significantly with treatment [99]. Appendix A lists the above mentioned accumulation biomarkers with the main characteristics.

#### 3.2.2. Response Biomarkers

There are different mediators of inflammation, oxidative stress and apoptosis, induced by the initial accumulation of a substrate. They are involved in different pathogenic processes of the disease and are related to both systemic involvement and specific tissue alterations. Several independent studies have focused on the assessment of these mediators as potential predictive markers of the disease and have been associated with both systemic vasculopathy and organ-specific alterations. The following describes those we consider most relevant, classified into three sub-groups: biomarkers related to systemic vasculopathy, biomarkers related to nephropathy, and biomarkers related to cardiomyopathy.

##### New Biomarkers Related to Systemic Vasculopathy

There is evidence that the exposure of normal dendritic and macrophage cells to Gb3 and DGJ (a-gal A inhibitor) results in an increase in the production of IL-1 and TNF-α cytokines [100]. In addition, bovine aortic endothelial cells treated with Gb3 increase the expression of factors involved in fibrosis, angiogenesis and apoptosis, i.e., TGF-1 **β**, VEGFR2, VEGF, FGF-2 and P-p38 [54]. Loss of a-gal A activity in human endothelial cell cultures has also been shown to produce a specific decrease in eNOS activity and a specific increase in 3-nitrotyrosine (3-NT) levels, considered as a marker of oxidative stress. These results were confirmed by demonstrating that 3-NT levels were significantly increased in plasma and in aortic tissue homogenate of knockout mice compared to wild-type [27].

Changes in circulating levels of mediators related to inflammation were also shown in patients with FD. Thus, increased levels of plasma sICAM-1, sVCAM-1, P-selectin, IL-6 and TNF-α [25,101], elevated expression levels of IL-1 and TNF-α in PBMC cells, and an overexpression of IL-6 and IL-1 in the monocyte subpopulation [100] were found in patients with the classic phenotype. Serum levels of myeloperoxidase (MPO), induced by pro-inflammatory status that contribute to the progression of vascular injury by inducing inflammatory and oxidative response, were also found to be increased in patients and were identified as a risk factor for the development of vascular events in males [102].

Alterations in oxidative status were also observed. Thus, in untreated patients with the classic phenotype, concentrations of plasma 3-NT were found to be significantly elevated [27], as well as an altered glutathione metabolism (assessed by erythrocyte levels of GSH and GPx activity), elevated levels of lipid peroxidation markers in plasma (TBARS, malondialdehyde), and elevated levels of urinary NO [103]. Another study targeting patients treated with ERT also showed changes in glutathione metabolism, in addition to elevated levels of plasma malondialdehyde and protein oxidation markers (carbonyl groups, and di-Tyr urinary levels). Some of these markers correlated with the levels of urinary Gb3 and/or plasma IL-6, suggesting that Gb3 could induce an inflammatory and pro-oxidant response [101].

##### New Biomarkers Related to Nephropathy

Lyso-Gb3 has been shown to induce, in cultured human podocytes, overexpression of TGF-β1 and CD74 [42], which are involved in the regulation of inflammation and development of renal fibrosis [51,52], overexpression of the ECM proteins fibronectin and type IV collagen [42], and increased Notch1 signaling, a pathway involved in the induction of podocyte injury and renal fibrosis, as well as overexpression of MCP-1, RANTES (Regulated on Activation, Normal T Cell Expressed and Secreted) and fibronectin. Inhibition of Notch1 signaling prevents the overexpression of such mediators [104]. Lyso-Gb3 is also able to increase the expression of CD80 [105] and the urokinase-type plasminogen activator receptor (uPAR) [106] in cultured human podocytes. CD80 plays an important role in regulating immune response [47,50], and can be expressed in podocytes and tubular cells in other glomerulopathies [48,49]. The uPAR is a transmembrane receptor that can participate in podocyte injury and detachment [107,108].

In cultured renal tubular cells, exposure to plasma lyso-Gb3 or Gb3 also results in an increase in TGF-β1 levels, changes in levels of epithelium-mesenchymal transition markers (decrease in E-caderin and increase in N-caderin and α-SMA), and an increase in levels of fibronectin and collagen IV [55]. Finally, a study with a Fabry mouse model showed that renal expression levels of TSP-1, TGF-β1, VEGF, VEGFR2 and FGF-2, related to angiogenesis and fibrosis, as well as that of P-p38 levels, and caspase-6 and caspase-9 activities, related to apoptosis, were increased compared to those of WT mice [54].

On the other hand, there are several studies in patients focused on the assessment of severity or predictive biomarkers. One study found in a group of patients that urinary excretion of CD80, podocyturia and proteinuria were significantly higher compared to control subjects, with no differences in levels of urinary CD80 or podocyturia between treated and untreated patients [105]. Another study in urine from patients found that the expression of uPAR in podocytes was higher in patients, and this was negatively associated with treatment [106]. Several studies have also proposed podocyturia as a potential early marker of renal injury [105,109,110,111]. However, studies focused on the evaluation of podocyturia provide poorly reproducible results. One of the possible causes is the lack of methodological standardization, which leads to an increase in the variability of data obtained between different studies, which make intercomparison of results difficult [112,113].

Urinary excretion of some podocyte proteins, including podocalyxin, podocin, nephrin or synaptopodin, has also been quantified as markers of podocyte injury. A recent study developed an LC-MS/MS quantification method based on cleavable stable isotopically labeled peptides to quantify levels of podocalyxin and podocin in urine in a cohort of FD patients [114], and showed that urinary levels of podocin may be a good diagnostic marker for untreated male patients with the classic phenotype, and were significantly associated with levels of urinary lyso-Gb3 in the total number of patients included in the study [115]. Several proteomic studies have also found potential early biomarkers [116]. On the one hand, in a study evaluating urine from pediatric male patients with normal renal function and without micro albuminuria, it was found that the concentration of prosaposin, a precursor protein of the activating proteins of some lysosomal hydrolases (saposins A, B, C and D), was significantly increased in pre-treated patients. These results were verified using LC-MS/MS, where increased levels of the GM2 activator protein [117] were also observed. Another study found that the excretion of some proteins was increased in patients, especially prostaglandin H2 D-isomerase, complement-c1q tumor necrosis factor-related protein (C1Q/TNF), and Ig kappa chain V-III, probably due to tubular dysfunction present in such patients. In addition, prostaglandin H2 D-isomerase had a different glycosylation pattern compared to the control subjects [118]. In the Matafora et al. study of patients in the early stage of the disease, uromodulin, involved in the pathogenesis of renal tubular fibrosis, prosaposin and prostaglandin H2 D-isomerase were validated as possible early markers [119]. It was also shown that some male patients had abnormal expression profiles of uromodulin, and that the degree of its renal expression is associated with the degree of storage of primary substrate [120]. A recently targeted proteomic assay focused on urine from Fabry patients revealed that 20 potential biomarkers related to lysosomal function, renal injury, cardiac injury and inflammatory pathways, could be useful in the diagnosis and monitoring of disease progression. In relation to renal injury, it was found that nephrin, a specific podocyte marker, could be a sensitive marker for presymptomatic renal involvement. Podocalyxin, Fibroblast Growth factor 23, Cubilin and Alpha-1-Microglobulin/Bikunin Precursor (AMBP) were also increased in patients with renal involvement. Some of these proteins also appear to be associated with the progression of the disease. [121]

Finally, reductions in levels of prosaposin, GM2 activator protein, uromodulin or prostaglandin H2 D-isomerase in urine were observed with treatment in patients in the early stages of the disease [117,119] as well as the normalization of abnormal uromodulin expression profiles in some of them [120].

Plasma and serum samples were also studied by proteomics to identify possible markers of Fabry disease [116].

##### New Biomarkers Related to Cardiomyopathy

Significant elevations in levels of serum lyso-Gb3, and of different cytokines and adhesion molecules (IL-6, IL-2, IL-1, TNF-α, MCP-1, sICAM-1, and sVCAM) have been observed in patients with cardiomyopathy-associated mutations [66]. Significantly increased concentration of plasma sphingosine 1-phosphate (S1P), a growth factor that promotes hypertrophy in cardiomyocytes [122], has also been found in male patients with the classic phenotype [64]. Another study targeting patients with the non-classic phenotype who were in follow-up found that the mean baseline plasma levels of procollagen type III amino-terminal propeptide (PIIINP), S1P, Lyso-Gb3, and urinary levels of alpha-1 collagen (I), (III), (VII), and alpha-3 collagen (V) were elevated. The fact that baseline plasma levels of high sensitive troponin T (hsTnT), ultra-hsTnT, Lyso-Gb3 and urinary excretion of fragments of collagen alpha-1 (I), (III), (VII) and alpha-3 (V) correlated with the degree of cardiac fibrosis determined by LGE, especially hsTnT and ultra-hsTnI, suggested that such markers could be useful as predictors of cardiac fibrosis [63]. In another study in patients with *GLA* IVS4 + 919G > A gene variant, it was observed by microarray analysis that IL-18, a pro-inflammatory cytokine related to heart disease and cardiac hypertrophy [123], was the most expressed marker in differentiated cardiomyocytes derived from patients with Fabry cardiomyopathy. These cardiomyocytes showed hypertrophy and an accumulation of Gb3, and the combination of enzyme treatment plus IL-18 neutralization slowed the progression of hypertrophy in them. In addition, circulating IL-18 levels in patients with left ventricular hypertrophy were significantly higher than hypertensive controls with ventricular hypertrophy [60].

On the other hand, in patients with advanced Fabry cardiomyopathy, histological and immunohistochemical evaluation of cardiomyocytes have shown an accumulation of glycosphingolipids and significant increases in the expression of oxidative stress markers, inducible nitric oxide synthase (iNOS) and nitrotyrosine (NT). In addition, some patients have positive nuclear staining for 8-hydroxy-2-deoxyguanosine (8-OHdG), which is also a marker of oxidative stress, and they showed a degree of apoptosis with protein nitration significantly increased. These results suggest that Gb3 accumulation produces an increase in oxidative stress in cardiomyocytes and in the degree of apoptosis, contributing directly to cardiac dysfunction [62]. Another study found that levels of the serum 8-OHdG were significantly increased in patients with cardiomyopathy [124].

Increases in plasma levels of some markers related to ECM turnover, such as MMP-9, were also found in patients with FD [78,125] in addition to increases in BNP, MR-proANP, MMP-2, TNF, TNFR1, TNFR2, IL-6, galectin-1, Lyso-Gb3 and its analogues, some of them being associated with the severity of heart disease assessed by the MSSI scores [78]. However, the association between MMP-9 and cardiomyopathy could not be proven subsequently [126]. On the other hand, in a study in which 66% of patients had cardiac fibrosis at the onset of the disease, baseline serum levels of PIIINP, procollagen type I carboxy-terminal propeptide (PIPP) and collagen type I carboxy-terminal telopeptide (CICT) were increased [127]. Finally, in a study that Aguiar et al. [128] conducted in a cohort of patients grouped by the severity of cardiomyopathy, serum levels of biomarkers involved in synthesis (PIPP) and degradation (CICT, MMP1, MMP2) of collagen type I adjusted for bone turnover were analyzed and found that adjusted PIPP levels were significantly increased in patients, including the asymptomatic, being higher in those with ventricular hypertrophy. Such increased levels in asymptomatic patients suggested that adjusted PIPP could be used as an early biomarker of cardiac dysfunction.

Some of the markers mentioned above have been associated with cardiac manifestations, such as endocardial fractional shortening, ventricular hypertrophy, ventricular dysfunction, maximal left-atrial size or degree of cardiac fibrosis [63,78,125,128], suggesting that mediators of inflammation, or ECM remodeling or regulation may contribute to the detection of cardiac injury, and be useful as indicators of left ventricular dysfunction.

Some studies show that changes in the levels of some biomarkers are associated with response to treatment. In this way, serum levels of lyso-Gb3 and IL-6, IL-2, IL-1β, TNF-α, MCP-1, sICAM-1, and sVCAM in patients with Fabry cardiomyopathy, were observed to decrease with enzymatic therapy, in parallel with the decrease in left ventricular mass and interventricular septal thickness [66]. The reduction of mean serum levels of 8-OHdG in the Chen et al. study, and the reduction of IL-18 serum levels in the Chien et al. study was also observed. The levels of both biomarkers changed in parallel to the decrease in the left ventricular mass and ventricular mass index [60,124]. However, the levels of some mediators were also increased in ERT treated patients compared to untreated [78]. This may be due to the fact that treated patients present more severe clinical manifestations compared to untreated patients.

Some of these mediators would also be useful as predictors of the progression of cardiac injury. Thus, positive correlation was shown between plasma levels of S1P and both carotid artery intima-media thickness and left ventricular mass, in a group of hemizygous patients with the classic phenotype [64]. On the other hand, in untreated patients with the non-classic phenotype under follow-up, plasma levels of lyso-Gb3, hsTnT, and PIIINP also changed in parallel with the progression of the degree of fibrosis [63]. In the study by Chien et al. [60] changes in both serum and expression IL-18 levels in cardiac biopsies were associated with the progression of hypertrophic cardiomyopathy.

However, other studies do not find such associations of markers with the progression of the injury. For example, a prospective study observed an increase in the extent of fibrosis (LGE) in a cohort of patients during follow-up, although no significant changes in serum levels of PIIINP, PIPP, and CICT were observed [127].

These results were interpreted that the markers related to the fibrotic process in FD could also reflect the progression of injury to fibrosis in other affected organs. However, the contribution of bone turnover to the increase in levels of such markers could bias associated studies. Thereby, in the study of Aguiar et al. [128] it was observed that adjusted PIPP levels were the better independent predictor of left ventricular mass, and that the adjusted PIPP:CICT ratio, an indicator of the balance between synthesis and degradation of type I collagen, showed good diagnostic accuracy to detect LGE fibrosis. Furthermore, diagnostic accuracy was higher when adding this ratio in a logistic model performed with the established predictors, both ventricular mass and NT-proBNP, compared to the model that included only the latter.

With regard to T1 and T2 mapping derived from a cardiac MRI [129,130,131,132,133,134,135,136] mentioned above, it has recently been found that circulating levels of both cTnI and lyso-Gb3 are associated with lower native T1 values in patients with FD. In addition, T1 values appear to be an earlier marker than Tn, since they are related to lipid deposition [136]. On the other hand, circulating levels of Tn increased in Fabry patients with LGE and are associated with increased T2 values [137]. These results are reinforced by the results of another independent study where elevated levels of cTnI were found in patients with myocardial fibrosis and active inflammation as measured by simultaneous cardiac PET/MRI imaging technique [138].

Appendix A lists the above mentioned response biomarkers with the main characteristics.

## 4. Future Lines of Work

Regarding accumulation markers, Gb3, Gb2 and lyso-Gb3 isoforms and/or analogues appear to be promising tools for detecting atypical variants in female Fabry patients, although further studies are required to validate their diagnostic value. To accomplish this, it would be convenient to conduct studies with a greater number of patients in which a histological examination is performed in parallel, in order to correctly classify the individuals, although we acknowledge that an invasive technique is not always feasible. To study their value as progression markers, further studies with patients in the early stages of the disease should be performed, studying the relationship with other possible markers related to the initiation of the pathogenic cascade. Finally, the expression level of CD77 could be a promising indicator of tissue accumulation, especially in renal and cardiac cells, and a good candidate for monitoring disease progression and treatment effectiveness in controlled studies.

The previously mentioned response markers, related to vasculopathy, nephropathy and/or cardiomyopathy, are involved in the pathogenic process ranging from the onset of the inflammatory reaction to the development of irreversible fibrosis. In general, they are not very specific markers. However, those that may have a more direct implication with the onset of renal or cardiac injury induced by the stored or exposed substrate, combined with the different clinical and accumulation markers, might be helpful in detecting kidney or heart injury in the early stages. These might include TGF-b1, CD80, podocalyxin, podocin, nephrin or uromodulin as nephropathy related biomarkers, and S1P or IL-18 as biomarkers more related to cardiomyopathy. To validate the response biomarkers, there must also be evidence of Gb3 or lyso-Gb3 deposits in tissues, whenever the clinical context of the patient allows it.

The standardization of laboratory methods is also necessary in the evaluation of all these biomarkers to ensure the intercomparison of results.

## 5. Conclusions

The efficacy of enzyme replacement therapy in FD patients is greater when initiated in the early stages of the disease. However, early diagnosis at this stage represents a clinical challenge, especially in patients with the non-classic phenotype or who are asymptomatic, which drives the search for new and non-invasive biomarkers that help the early detection and prediction of disease progression. Throughout this review, several mediators have been cited, which have been studied as potential predictive markers of FD. These are directly or indirectly involved in the accumulation of a substrate in the various affected organs, or with the local and systemic response derived from it and have been related to the progression of the injury and the main clinical manifestations of the disease.

The incorporation of the mentioned markers in clinical practice could provide diagnostic value, increasing the current predictive detection ability of the established clinical, imaging, histological and biochemical diagnostic markers, especially in patients still asymptomatic or in the very early stages of the disease.

The lack of associated evidence in some studies between some accumulation or response biomarkers assessed and the different clinical manifestations or clinical response reflects the heterogeneity of the disease, i.e., the multiple mechanisms and mediators that contribute directly or indirectly, and independently, to the development of the different phenotypes observed. Moreover, the different designs of the studies (i.e., composition and size of the cohorts, inclusion criteria of the participants) that evaluate these relationships hamper a direct comparability. In this way, better-designed systematic and stratified follow-up studies are needed to validate these markers as predictive and/or prognostic indicators of the disease and their usefulness in clinical practice.

## Figures and Tables

**Figure 1 jcm-10-01664-f001:**
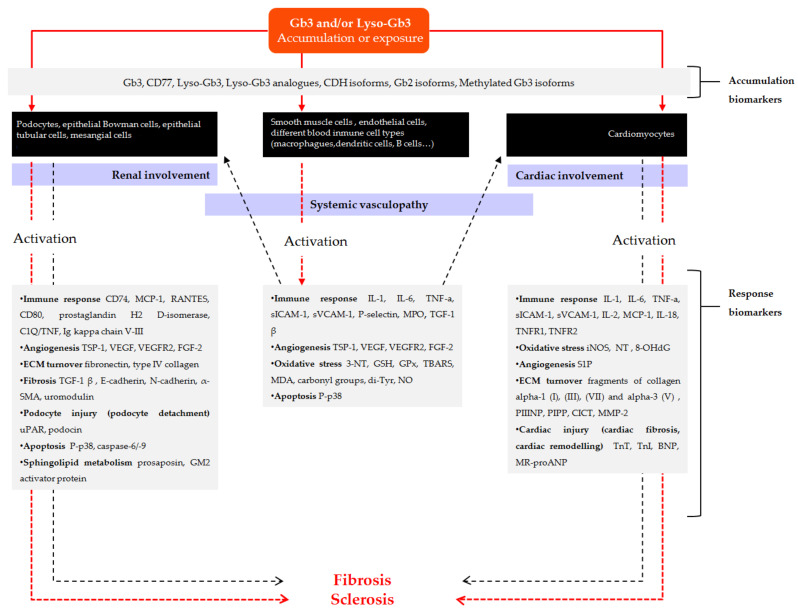
Scheme of the main pathogenic pathways with biomarkers involved in Fabry disease. Pathogenic mechanisms described in the development of vascular involvement are initiated from Globotriaosylceramide (Gb3) and/or Globotriaosylsphingosine (lyso-Gb3) accumulation and/or chronic exposure to smooth muscle cells, endothelial cells or different blood immune cell types (solid red arrow), which in turn initiates different pathogenic cascades that contribute to the progression of vasculopathy (dashed red arrow). Inflammation derived from systemic vasculopathy may also be involved in the progression of disease related to cardiomyopathy and nephropathy (dashed black arrows). In addition, substrate can also accumulate or deposit on other cell types such us podocytes, other renal cells, or cardiomyocytes (solid red arrows), inducing different pathogenic mechanisms that may contribute to the development of renal and cardiac fibrosis. Gray boxes include different substrates related to the accumulation or exposure, as well as mediators involved in different pathways of response to the initial injury and progression towards fibrosis, which have been identified in clinical and experimental studies and that can be useful as diagnostic or follow-up biomarkers. α-SMA: α-smooth muscle actin; BNP: brain natriuretic peptide; C1Q/TNF: complement-c1q tumor necrosis factor-related protein; CICT: collagen type I carboxy-terminal telopeptide; CDH: ceramide dihexoside; FGF-2: fibroblast growth factor-2; Gb2: Galabiosylceramide; Gb3: Globotriaosylceramide; GSH: glutathione; GPx: glutathione peroxidase; hsTn: high sensitive Troponin; IL: interleukin; iNOS: inducible nitric oxide synthase; lyso-Gb3: Globotriaosylsphingosine; MCP-1: monocyte chemo-attractant protein; MDA: malondialdehyde; MMP-2: matrix metalloproteinase-2; MPO: myeloperoxidase; MR-proANP: midregional pro-atrial natriuretic peptide; NO: nitric oxide; NT: nitrotyrosine; 8-OHdG: 8-hydroxydeoxyguanosine; PIPP: procollagen type I carboxy-terminal propeptide; PIIINP: procollagen type III amino-terminal propeptide; sICAM-1: soluble intercellular adhesion molecule 1; sVCAM-1: soluble vascular cell adhesion molecule 1; S1P: sphingosine-1- phosphate; TBARS: thiobarbituric acid reactive species; TGFβ-1: transforming growth factor beta-1; Tn: troponin; TNF-α: tumor necrosis factor alpha; TNFR: tumor necrosis factor receptor; TSP-1: thrombospondin-1; Tyr: tyrosine; uPAR: urokinase-type plasminogen activator receptor; VEGF: vascular endothelial growth factor; VEGFR: vascular endothelial growth factor receptor.

**Figure 2 jcm-10-01664-f002:**
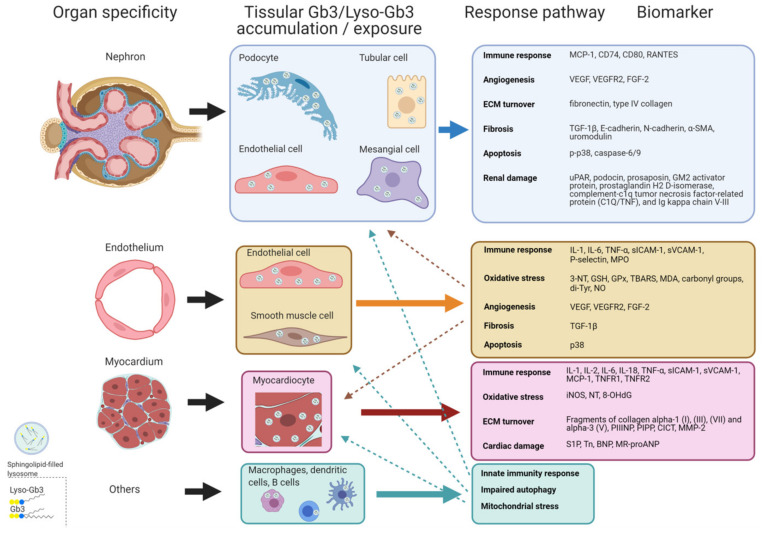
Scheme of potential pathogenic response pathways. Response pathways promoted from Gb3 and/or lyso-Gb3 accumulation and/or exposure that can develop in Fabry disease grouped by organ/tissue specificity, with some biomarkers involved. Dashed arrows indicate associations between systemic vasculopathy and renal and cardiac involvement. α-SMA: α-smooth muscle actin; BNP: brain natriuretic peptide; CICT: collagen type I carboxy-terminal telopeptide; FGF-2: fibroblast growth factor-2; Gb3: Globotriaosylceramide; GSH: glutathione; GPx: glutathione peroxidase; Tn: high sensitive Troponin; IL: interleukin; iNOS: inducible nitric oxide synthase; lyso-Gb3: Globotriaosylsphingosine; MCP-1: monocyte chemo-attractant protein; MDA: malondialdehyde; MMP-2: matrix metalloproteinase-2; MPO: myeloperoxidase; MR-proANP: midregional pro-atrial natriuretic peptide; NO: nitric oxide; NT: nitrotyrosine; 8-OHdG: 8-hydroxydeoxyguanosine; PIPP: procollagen type I carboxy-terminal propeptide; PIIINP: procollagen type III amino-terminal propeptide; sICAM-1: soluble intercellular adhesion molecule 1; sVCAM-1: soluble vascular cell adhesion molecule 1; S1P: Sphingosine-1- phosphate; TBARS: thiobarbituric acid reactive species; TGFβ-1: transforming growth factor beta-1; TNF-α: tumor necrosis factor alpha; TNFR: tumor necrosis factor receptor; Tyr: tyrosine; uPAR: urokinase-type plasminogen activator receptor; VEGF: vascular endothelial growth factor; VEGFR: vascular endothelial growth factor receptor.

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
