# Peer review of "Biomarkers in Fabry Disease. Implications for Clinical Diagnosis and Follow-up"

_jcm, 2021, doi:10.3390/jcm10081664_

Round 1

Reviewer 1 Report

This review sheds some light in the recent puzzling and sometimes conflicting scientific data on the search for new biomarkers in Fabry disease. The manuscript overviews the whole span from clinical papers to experimental studies, discussing the results with unbiased criticism.

The classification of new and known biomarkers needs some consistency. The definitions used in the text are in my view too many and sometimes confusing: primary, secondary, new, classic, early, predictive, accumulation and response biomarkers... Particularly in the abstract the definition of accumulation, response and primary biomarkers is not clear (line 40-42).

In addition  the term response biomarker can be misleading when dealing with the response to treatment or to vasculopathy, nephropahty of cardiomyopathy.

Instead of Classic Markers use Clinical Markers (line 162)

Line 56: the estimated incidence is quite updated, but indeed there are no recent published data, therefore a sentence stating that the estimated incidence could be underestimated is mandatory

Legend to Figure 1: line 152 153, LGE LVH, LVM  and LVMI are not shown in Fig 1... and  MR-proANP (line 156) not fully explained.

Line 214 to 224: this paragraph on the response to treatment could be placed at the end of this chapter (line 277).

Line 358: in addition to the proteomic studies reported in this section, the review by Rossi F et al. Molecular Genetics and Metabolism 132 (2021) 86–93 could added to the references

Line 464 to 471 deals with cMRI, i.e. clinical diagnostic biomarkers, and should be placed in the Classic (or better Clinical) Markers section, where the role of T1 mapping is already discussed (line 170). In addition the paper of Camporeale A et al., Prognostic Impact of T1 Lowering in Fabry Disease, Cardiovasc Imaging 2019, could be added to the references.

Author Response

Dear reviewer,

We appreciate your participation in reviewing this manuscript, as well as all the commentaries you have provided on it. Here we detail our answers and the changes in the manuscript according to your suggestions:

  1. The definitions used in the text are in my view too many and sometimes confusing: primary, secondary, new, classic, early, predictive, accumulation and response biomarkers... 

Answer:

We have simplified the definitions of biomarkers. There will be only 2 categories:

- Accumulation biomarkers include substrate of the primary blocked pathway, globotriaosylceramide and its derivatives.

- Response biomarkers include all reactive mediators related to injury in the different pathogenic pathways.

The definition is explained in the following sentence: “Finally, the biomarkers found in the filtered results have been classified into two groups, biomarkers related to the accumulation of substrate and those related to the response to the initial injury.” (Introduction: page 2, highlighted lines)

We assume that both accumulation and response biomarkers could be potentially used as indicators of response to treatment.

  1. Instead of Classic Markers use Clinical Markers (line 162).

Answer:

We have substituted the term classic for clinical biomarkers throughout the text.

  1. Line 56: the estimated incidence is quite updated, but indeed there are no recent published data, therefore a sentence stating that the estimated incidence could be underestimated is mandatory.

Answer:

We have updated the sentence according to your suggestion

  1. Legend to Figure 1: line 152 153, LGE LVH, LVM and LVMI are not shown in Fig 1... and MR-proANP (line 156) not fully explained.

Answer:

Sorry, there was a mistake as part of the legend had been copied from another previous table. Now we have corrected it. Abbreviation of MR-pro ANP is also corrected in  the legend of supplementary table S2

  1. Line 214 to 224: this paragraph on the response to treatment could be placed at the end of this chapter (line 277).

Answer:

The structure of this section was devised in the following way: firstly referring evidences of Lyso-Gb3 in the following order: as detection marker, as its association with clinical manifestations, and association with response to treatment; secondly the same items for the Lyso-Gb3 analogues as well as Gb3/Gb2 isoforms. 

We hope you don’t mind if we leave it as it is. Thank you for understanding.

  1. Line 358: in addition to the proteomic studies reported in this section, the review by Rossi F et al. Molecular Genetics and Metabolism 132 (2021) 86–93 could be added to the references

Answer: We have added this reference

  1. Line 464 to 471 deals with cMRI, i.e. clinical diagnostic biomarkers, and should be placed in the Classic (or better Clinical) Markers section, where the role of T1 mapping is already discussed (line 170). In addition the paper of Camporeale A et al., Prognostic Impact of T1 Lowering in Fabry Disease, Cardiovasc Imaging 2019, could be added to the references.

Answer:

In this paragraph we cited some references (126-133] of T1 and T2 as imaging markers of cardiomyopathy in general, not related to Fabry disease (this is the reason to not be included it in the above Clinical Markers of Fabry disease section).

The following explanation concerns associations of some biological biomarkers with those of cardiac imaging ones. Nevertheless, we have rewritten the sentence according to your suggestion, with the aim that this is a cardiac section.

If you keep thinking it generates confusion we can remove the following sentence (with its references): “With regard to T1 and T2 mapping derived from cardiac MRI [126-133]”...

We have also added the suggested reference in 3.1 Clinical Markers section.

In addition to the changes made from your suggestions, you can see that there are other changes we have made in the manuscript due to:

- Suggestions proposed by the other reviewer

-Corrections of grammatical mistakes detected in Figure 1. Also, we have added some abbreviations that were missing in the legend.

Reviewer 2 Report

This paper is a well-written scholarly summary of current status of biomarkers for Fabry disease, and I congratulate the authors on taking a thoughtful organ-specific approach. Mechanistically there is great variability in the strength of the disease association between many of these molecules and phenotype, and I would like to see:

  1. the potential pathways developed further, perhaps with use of figures to increase readability
  2. clearer indication of the strength of associations
  3. possible links to fibrotic pathways, as that will define the point of irreversibility

However, the manuscript is of high interest to clinicians and scientists working in the field, and provides a useful backdrop to hypothesis generation. 

Author Response

Dear reviewer,

We appreciate your participation in reviewing this manuscript, as well as all the commentaries you have provided on it. Here we detail our answers and the changes in the manuscript according to your suggestions:

This paper is a well-written scholarly summary of current status of biomarkers for Fabry disease, and I congratulate the authors on taking a thoughtful organ-specific approach. Mechanistically there is great variability in the strength of the disease association between many of these molecules and phenotype, and I would like to see:

  1. the potential pathways developed further, perhaps with use of figures to increase readability
  2. clearer indication of the strength of associations
  3. possible links to fibrotic pathways, as that will define the point of irreversibility

However, the manuscript is of high interest to clinicians and scientists working in the field, and provides a useful backdrop to hypothesis generation. 

Answer:

We have tried to address your indications by adding a new figure indicating the main pathophysiological processes explained in the manuscript. We hope it fits your request. But if you prefer to detail other aspects in the figure, we can review it again.

We consider that in the Pathogenesis section (page 3, third paragraph, lighted lines) we refer to the development of irreversible fibrosis, with 7 references. Also in section 3.2.2.2. “New biomarkers related to nephropathy”, the association of renal fibrosis and specific biomarkers are more detailed, as well as in table S2

Nevertheless, we have rewrited this sentence in section 2. Pathogenesis pointing that it is a development towards irreversible fibrosis.

In addition to the changes made from your suggestions, you can see that there are other changes we have made in the manuscript due to:

- Suggestions proposed by the other reviewer

-Corrections of grammatical mistakes detected in Figure 1. Also, we have added some abbreviations that were missing in the legend.